# Isolation of the Thermostable $\beta$-Glucosidase-Secreting Strain *Bacillus altitudinis* JYY-02 and Its Application in the Production of Gardenia Blue

Jingyuan Yang,[a] Chao Wang,[a] Qunqun Guo,[a] Wenjun Deng,[a] Guicai Du,[a] Ronggui Li[a]

[a]College of Life Sciences, Qingdao University, Qingdao, P.R. China

**ABSTRACT** Gardenia blue (GB) is a natural blue pigment widely used in textiles and the pharmaceutical industry. The geniposide in gardenia fruits can be hydrolyzed by $\beta$-glucosidase to form genipin, which reacts with amino acids to produce GB. In this study, a bacterial strain which secreted thermostable $\beta$-glucosidase (EC 3.2.1.21) was isolated from soil and identified as *Bacillus altitudinis* JYY-02. This strain could potentially be used for GB production from geniposide by fermentation. Optimal fermentation results were achieved at pH 6.5 or 8.0 at 45℃ for 45 h with additional sucrose. To obtain a large amount of $\beta$-glucosidase, the whole genome of *B. altitudinis* JYY-02 was sequenced and annotated; it is 3,727,518 bp long and contains 3,832 genes. The gene encoding $\beta$-glucosidase (*bgl*) in *B. altitudinis* JYY-02 was screened from the genome and overexpressed in *Escherichia coli* BL21(DE3). The recombinant $\beta$-glucosidase was purified by affinity chromatography on a Ni Sepharose 6 fast flow (FF) column. The optimal temperature, pH, and $K_m$ values for the recombinant $\beta$-glucosidase were 60℃, pH 5.6, and 0.331 mM, respectively, when p-nitrophenyl-$\beta$-D-glucopyranoside (pNPG) was used as the substrate. The recombinant $\beta$-glucosidase catalyzed the deglycosylation reaction of geniposide, which was then used to produce GB.

**IMPORTANCE** $\beta$-Glucosidases are enzymes capable of hydrolyzing $\beta$-glucosidic linkages present in saccharides and glycosides and have many agricultural and industrial applications. Although they are found in all domains of living organisms, commercial $\beta$-glucosidases are still expensive, limiting their application in industry. In the present study, a thermostable $\beta$-glucosidase-producing strain was obtained for GB production by fermentation, engineered bacteria were constructed for preparing recombinant $\beta$-glucosidase, and a one-step method to purify the recombinant enzyme was established. A large amount of purified $\beta$-glucosidase was easily obtained from the engineered bacteria for industrial applications such as GB production.

**KEYWORDS** *Bacillus altitudinis* JYY-02, $\beta$-glucosidase, genome, gardenia blue pigment, fermentation, enzymatic characterization

In terms of natural pigments, blue is valuable due to its rarity in natural sources (1). Gardenia blue (GB) is one of the major sources of natural blue pigment (2). It has been widely used in textiles and the medical field because of its outstanding properties, including strong coloring power, high stability, environmental friendliness, nontoxicity, and high biological activity. Denim, silk, and other fabrics can be dyed with GB instead of indigo (3). Compared to indigo blue, the environmental benefit and stability are the obvious advantages of GB. In addition to its use in the traditional dyeing industry, GB has been developed as a dental plaque stain (4). Due to its unique biological activity, GB has been found to have excellent anti-inflammatory and antidepressant effects (5, 6).

Commercial GB comes from the glucoside (geniposide) in gardenia fruit. When catalyzed by $\beta$-glucosidase, the hydrolysis of one molecule of geniposide produces one molecule of genipin and one molecule of glucose. Genipin can further react with amino acids to generate

Address correspondence to Guicai Du, duguicai@qdu.edu.cn, or Ronggui Li, lrg@qdu.edu.cn.

The authors declare no conflict of interest.

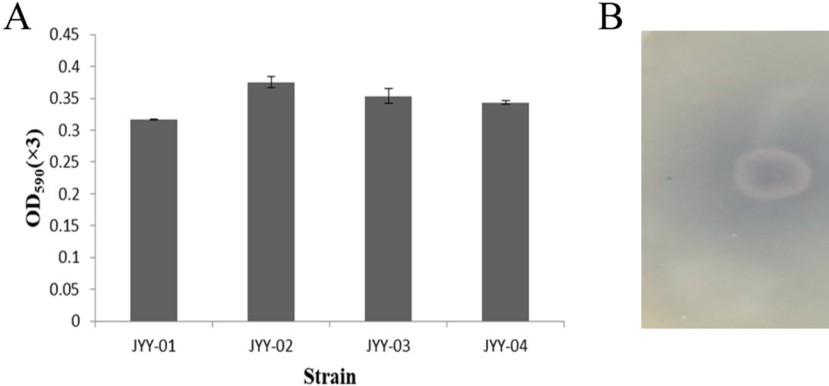

**FIG 1** Screening of bacterial strains for GB production. (A) Abilities of four different strains to produce gardenia blue by fermentation. (B) *B. altitudinis* JYY-02-produced GB on geniposide-glycine screening medium.

GB (7, 8). During the process of gardenia yellow pigment extraction in industry, a large amount of geniposide is discarded as an impurity (9, 10). The reuse of geniposide-containing waste liquid for GB production can effectively reduce the waste of resources and contribute to the circular economy.

The commonly used methods for production of GB include a fermentation method and a two-step method. The former is carried out by the hydrolysis of geniposide and the synthesis of GB simultaneously through microbial fermentation. However, the drawbacks of this method lie in the low purity of the GB and its limited yield, depending on the microbial growth conditions. For the two-step method, geniposide is first hydrolyzed to genipin by β-glucosidase, followed by the reaction of genipin with amino acids to form GB (see Fig. S1 in the supplemental material). This method provides the GB with a higher level of purity but at a higher cost as well (11). Glycine, alanine, valine, leucine, and other amino acids can be used to prepare GB, and glycine is one of the most readily available amino acids in molecular laboratories (11, 12).

β-Glucosidase is the key rate-limiting factor in the process of the hydrolysis of geniposide. It is mainly derived from microorganisms such as bacteria, *Aspergillus* spp., and yeasts. The traditional method for extraction of β-glucosidase from microorganisms has the disadvantages of being time-consuming and tedious. With the development of modern molecular biology, heterologous expression of the gene coding for β-glucosidase was applied to produce this recombinant protein, in which a large amount of active enzymes can be accumulated in the engineered bacteria. This method shows the obvious advantages of high yields and a simple purification process. Currently, temperature is one of the major limitations on the production efficiency of GB (11). The reaction process of genipin with amino acids requires a high temperature, above 90°C; however, the lack of heat-resistant β-glucosidase limits the effective combination of two-step reactions (11). β-Glucosidase with thermal stability is the key to solving this problem. *Thermotoga maritima* is one of the most intensively thermophilic bacteria that produce thermostable β-glucosidase. It can only grow optimally in submarine craters, under conditions which cannot be replicated for industrial production (13, 14).

In this study, *Bacillus altitudinis* JYY-02 producing thermostable β-glucosidase with an optimal temperature of 60°C was isolated from soil. The fermentation conditions for the bacteria to produce GB were characterized. Separately, the gene coding for β-glucosidase was overexpressed in *Escherichia coli*, and active recombinant β-glucosidase was obtained. The findings in this study provide a new method for the preparation of β-glucosidase for production of GB.

## RESULTS

**Identification of strains.** Four bacterial strains with blue colonies and surrounding background were screened out on geniposide-glycine screening medium. A strain which could produce the highest content of GB under the same fermentation conditions was selected for further study (Fig. 1). The isolate was Gram positive with endospores, and no flagellum was observed (see Fig. S2 in the supplemental material). The physiological and

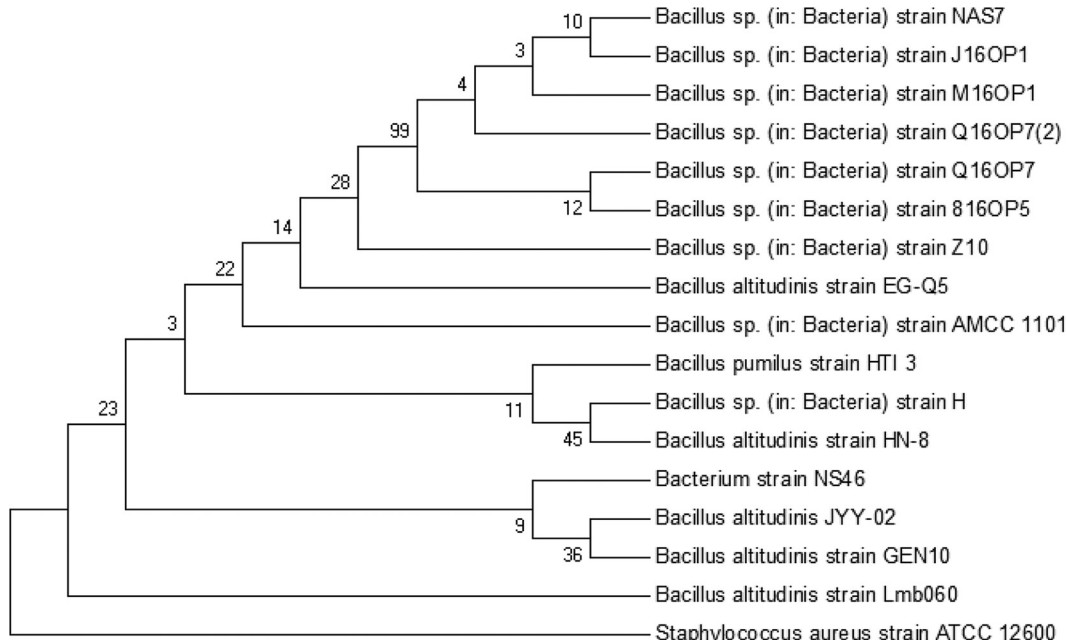

**FIG 2** Phylogenetic tree constructed from the 16S rRNA gene sequences of *B. altitudinis* JYY-02 and other species.

biochemical characteristics (summarized in Table S1) showed that it is a strain belonging to the genus *Bacillus*, according to *Bergey's Manual of Systematic Bacteriology* (15). To further identify this isolate, the 16S rRNA gene was amplified by PCR and sequenced, and a BLAST search of the NCBI database verified that it was homologous to that of *Bacillus altitudinis*. Therefore, the strain was identified as *Bacillus altitudinis* JYY-02 (Fig. 2).

**Conditional optimization for fermentation of GB.** To determine the optimal conditions for GB production through fermentation of *B. altitudinis* JYY-02, the effect of culturing time on GB synthesis was investigated. The results showed that the content of GB would accumulate gradually with extension of the culturing time but would tend to stabilize after 45 h (Fig. 3A). In the one-way experiment, the initial pH values had an influence on the GB production; two optimal pH values of 6.5 and 8.0 were observed (Fig. 3B), which might be because one pH was suitable for *β*-glucosidase and the other was optimal for the growth of *B. altitudinis* JYY-02. Further experimentation verified that pH 8 was the optimal pH for the growth of *B. altitudinis* JYY-02 (Fig. S3). The optimal temperature for producing blue pigment determined by the single-factor experiment was 45°C (Fig. 3C). The addition of different carbon sources to the medium was found to increase the GB to different extents, and sucrose showed the most obvious promotion of GB production (Fig. 3D). Analysis by orthogonal experiment indicated that additional carbon sources had the greatest effect on the fermentation results, followed by the fermentation temperature. The maximum amount of GB was obtained by fermentation at an initial pH of 8.0 and a temperature of 45°C for 45 h supplied with additional sucrose (Tables 1 and 2).

**Whole-genome sequencing of *B. altitudinis* JYY-02.** Whole-genome sequencing of *B. altitudinis* JYY-02 showed that the genome size was 3,727,518 bp, with a total of 3,832 genes. The average length of the genes was 856.89 bp, and the GC content of the whole genome was 41.91%. The distribution of each element is displayed on the gene circle based on the results of gene distribution, noncoding RNA (ncRNA) distribution, annotation, and other data in Fig. 4. Comparison of the genome with the Cluster of Orthologous Groups of proteins (COG) database revealed that the largest number of genes (326) were related to amino acid transport and metabolism, followed by those related to transcription. However, only 1 gene was related to RNA processing modifications, chromatin, and kinetics (Fig. S4). Five noncoding RNAs were present in the genome of this strain, which together accounted for 1.21% of the genome size. Analysis of the genome with several databases, such as the Gene Ontology (GO) database (Fig. S5) and the Kyoto Encyclopedia of Genes and Genomes

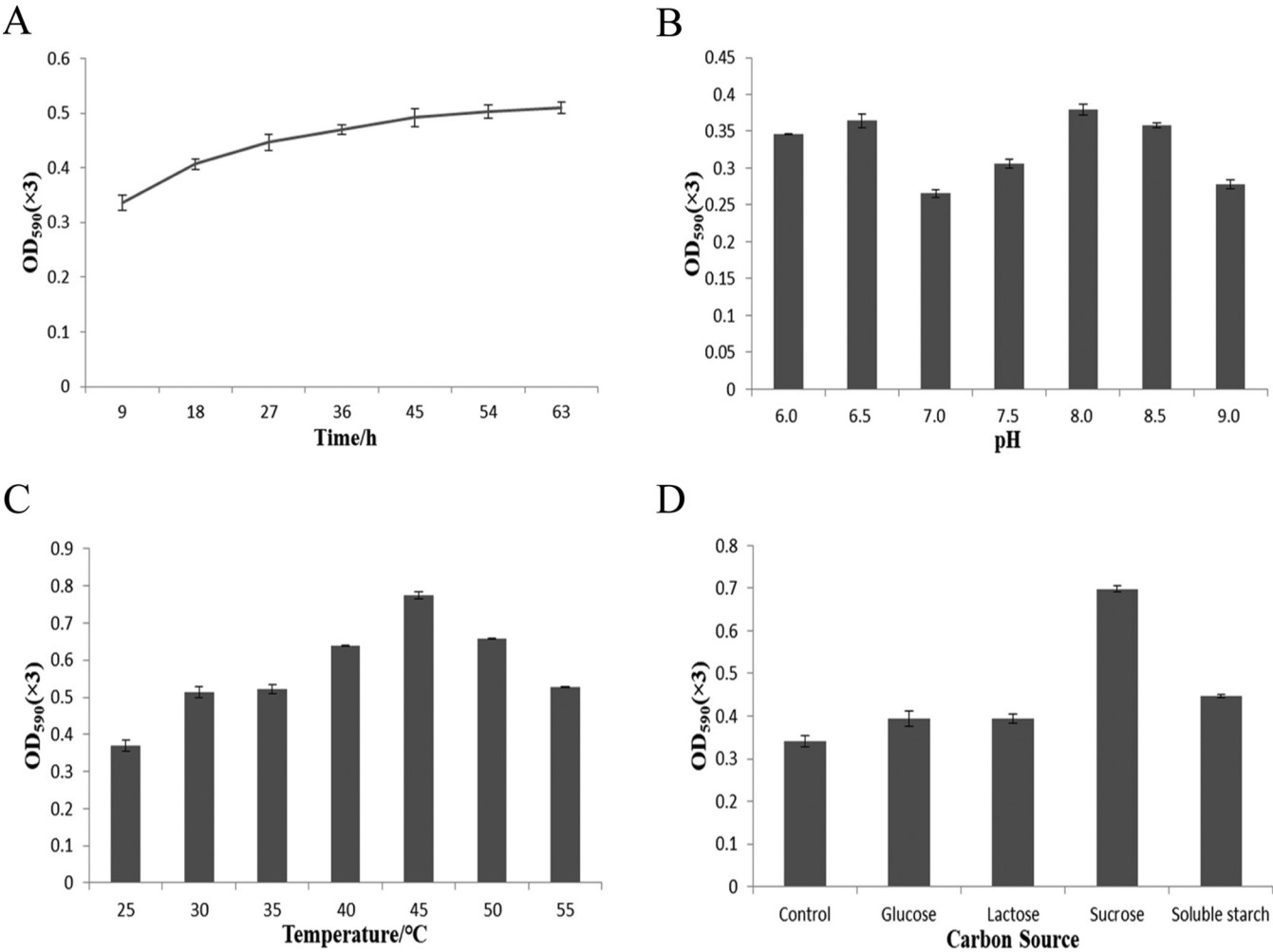

**FIG 3** Graphs showing the factors influencing GB production by fermentation of *B. altitudinis* JYY-02. Effect on production of time (A), pH (B), temperature (C), and the addition of an additional carbon source (D).

(KEGG) database (Fig. S6), revealed that there were several genes related to drug resistance and hydrolase activity.

**Preparation of the recombinant β-glucosidase.** The sequence encoding β-glucosidase *B. altitudinis* JYY-02 was screened out by annotation of the genomic sequence and a BLAST search of the predicted sequence against those encoding β-glucosidase in the NCBI database. According to the gene sequenced, a pair of primers were designed, and the gene for β-glucosidase was amplified by PCR from the genomic DNA of *B. altitudinis* JYY-02. Sequence analysis indicated that the open reading frame (ORF) of the PCR product was 1,467 bp, encoding a peptide of 488 amino acid residues (Fig. S7). DNA sequence analysis using BLAST showed that the *bgl* gene of *B. altitudinis* JYY-02 was 100.00% and 99.93% homologous to that of *Bacillus altitudinis* and *Bacillus pumilus*, respectively. A BLAST search of the amino acid sequences indicated that the β-glucosidase of *B. altitudinis* JYY-02 was 99.8%, 95.9%, and 95.9% homologous to that of *B. altitudinis*, *B. pumilus*, and *Bacillus safensis*, respectively (Fig. S8). Protein structure prediction showed that the β-glucosidase of *B. altitudinis* JYY-02 contained 39.96% α-helix, 9.22% β-turn, 16.80% extension chain, and 34.02% random curl (Fig. 5). The abundance of α-helix means that the protein might have higher stability.

The *bgl* gene amplified by PCR was cloned into pET-15b to construct the recombinant vector pET-15b-*bgl* and transformed into *E. coli* BL21(DE3) to construct the engineered bacteria. The recombinant β-glucosidase, with a relative molecular weight of 54 kDa, accumulated in the engineered bacteria after induction by isopropyl-β-D-thiogalactopyranoside (IPTG) and was easily purified to homogeneity by affinity chromatography on a nickel

**TABLE 1** Design table for orthogonal experiments

| Factor | Time (h) | Temp (°C) | Initial pH | Carbon source |
|---|---|---|---|---|
| 1 | 36 | 40 | 6.5 | Sucrose |
| 2 | 36 | 45 | 7.5 | Soluble starch |
| 3 | 36 | 50 | 8.0 | Glucose |
| 4 | 45 | 40 | 7.5 | Glucose |
| 5 | 45 | 45 | 8.0 | Sucrose |
| 6 | 45 | 50 | 6.5 | Soluble starch |
| 7 | 54 | 40 | 8.0 | Soluble starch |
| 8 | 54 | 45 | 6.5 | Glucose |
| 9 | 54 | 50 | 7.5 | Sucrose |

column (Fig. 6). The recovery yield of β-glucosidase was 71.75% after the one-step purification by affinity chromatography (Table 3).

**Characteristics of the recombinant β-glucosidase.** An activity assay of the recombinant β-glucosidase showed that the optimal temperature was 60°C (Fig. 7A), and the optimal pH value was pH 5.6 (Fig. 7B). Ions had different effects on the activity of β-glucosidase; $Zn^{2+}$ significantly enhanced its activity, while $NH_4^+$, $Ni^{2+}$, and $Mg^{2+}$ had a slightly inhibiting effect, and $K^+$ had no obvious effect on the activity (Fig. 7C). Thermal-stability tests showed that more than 80% of the activity of the recombinant β-glucosidase could be preserved when it was treated at 70°C for 1.5 h, and the residual activity was still at 63.078% even after being treated at 100°C for 1.5 h, which indicated that the recombinant β-glucosidase was a thermostable protein (Fig. 7D). According to the Lineweaver-Burk plot, the $V_{max}$ and $K_m$ values (kinetic parameters) of the recombinant β-glucosidase were determined to be 19.194 U·min$^{-1}$·mg$^{-1}$ and 0.332 mM, respectively, with p-nitrophenyl-β-D-glucopyranoside (pNPG) as the substrate (Fig. 8).

**GB preparation with the recombinant β-glucosidase.** To investigate whether the recombinant β-glucosidase could be used for GB production, gardenia glycoside was hydrolyzed by the recombinant β-glucosidase and then incubated with glycine. The results showed that a blue product formed with the maximum absorption at 590 nm identical to that of GB, and the yield of GB increased with an increase in the recombinant β-glucosidase along with a darkening of the blue color, which indicated that the recombinant β-glucosidase had the ability to hydrolyze gardenia glycoside for production of GB (Fig. 9).

## DISCUSSION

The β-glucosidases are found in all domains of living organisms, where they play essential roles in the removal of nonreducing terminal glucosyl residues from saccharides and glycosides, which are related to many physiological processes, such as glycoside metabolism, defense, cell wall lignification, and biomass conversion (16). β-Glucosidases are widely applied in the food, animal feed, pharmacy, and textile industries. Several *Bacillus* species, such as *B. subtilis* (17) and *B. stratosphericus* (18), have been found to secrete β-glucosidases, but

**TABLE 2** Visual analysis of orthogonal experiments

| Factor | Time (h) | Temp (°C) | Initial pH | Carbon source | Result |
|---|---|---|---|---|---|
| 1 | 1 | 1 | 1 | 1 | 0.8705 |
| 2 | 1 | 2 | 2 | 2 | 0.4707 |
| 3 | 1 | 3 | 3 | 3 | 0.497 |
| 4 | 2 | 1 | 2 | 3 | 0.4675 |
| 5 | 2 | 2 | 3 | 1 | 1.3285 |
| 6 | 2 | 3 | 1 | 2 | 0.531 |
| 7 | 3 | 1 | 3 | 2 | 0.5175 |
| 8 | 3 | 2 | 1 | 3 | 0.5705 |
| 9 | 3 | 3 | 2 | 1 | 1.046 |
| k1 | 0.613 | 0.619 | 0.657 | 1.082 | |
| k2 | 0.776 | 0.790 | 0.661 | 0.506 | |
| k3 | 0.711 | 0.691 | 0.781 | 0.512 | |
| R | 0.163 | 0.171 | 0.124 | 0.576 | |

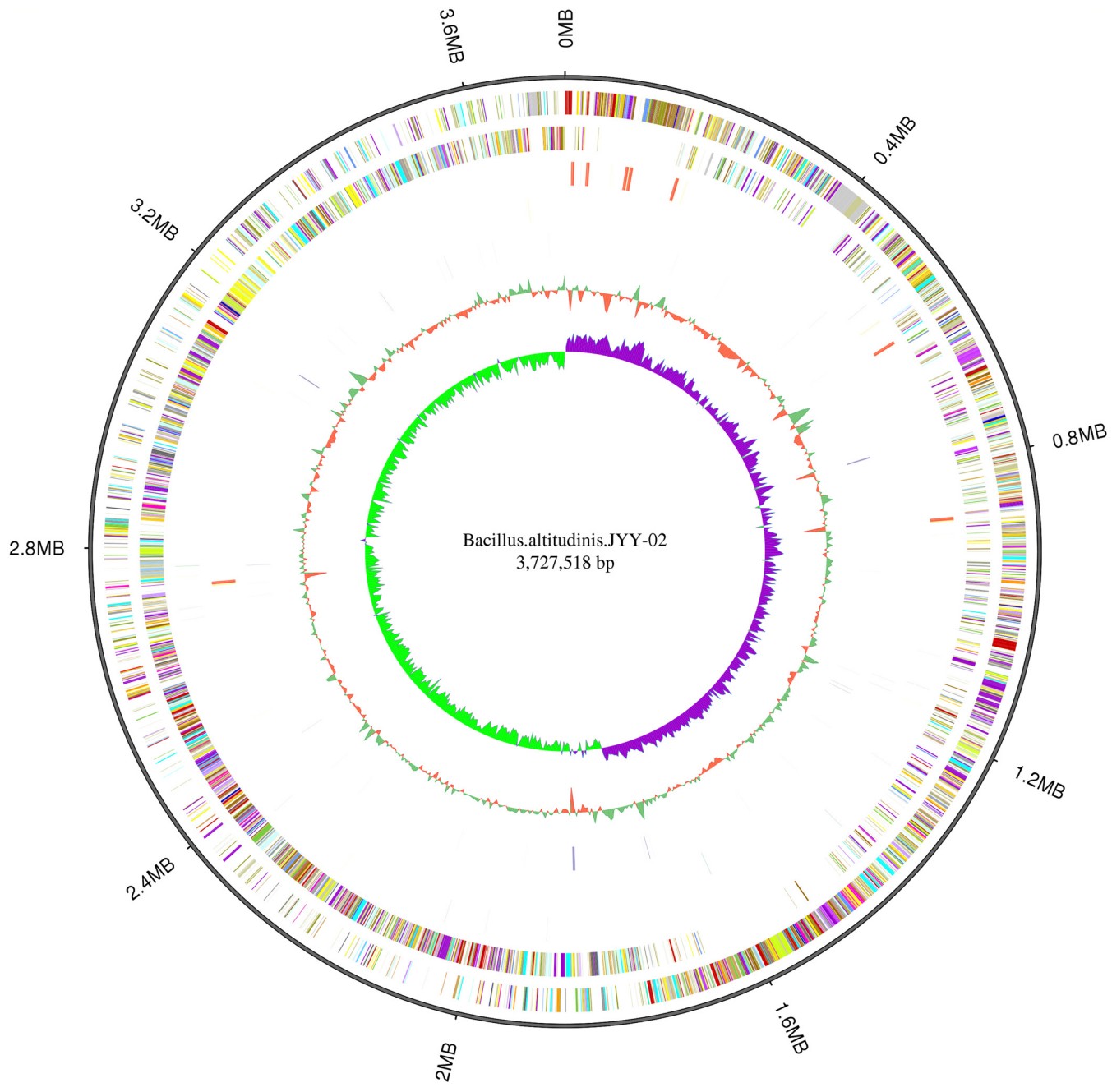

**FIG 4** Genome circles representing (outer to inner) genome size; forward strand gene, colored according to the Cluster of Orthologous Groups (COG) classification; reverse strand gene, colored according to the COG classification; forward-strand ncRNA; reverse-strand ncRNA; repeat; GC; GC skew.

the enzyme yield still needs to be improved. The fungus *Trichoderma reesei* was also reported to produce β-glucosidase but in very small amounts (19). In this study, one strain from the genus *Bacillus* was isolated and identified as *Bacillus altitudinis* JYY-02 in order to screen a suitable isolate for GB production. The strain was found to be feasible for GB production by fermentation, and the optimal conditions were determined by an orthogonal experiment. However, we found that it was very difficult to purify the β-glucosidase of *B. altitudinis* JYY-02 participating in GB synthesis due to its rapid degradation. Supplementation of proteinase inhibitors such as EDTA and PMSF (phenylmethylsulfonyl fluoride) had no obvious effect on the stability of the β-glucosidase.

To obtain enough β-glucosidase to study its properties and the synthesis of GB, the genome of *B. altitudinis* JYY-02 was sequenced, and a gene encoding β-glucosidase, which

A

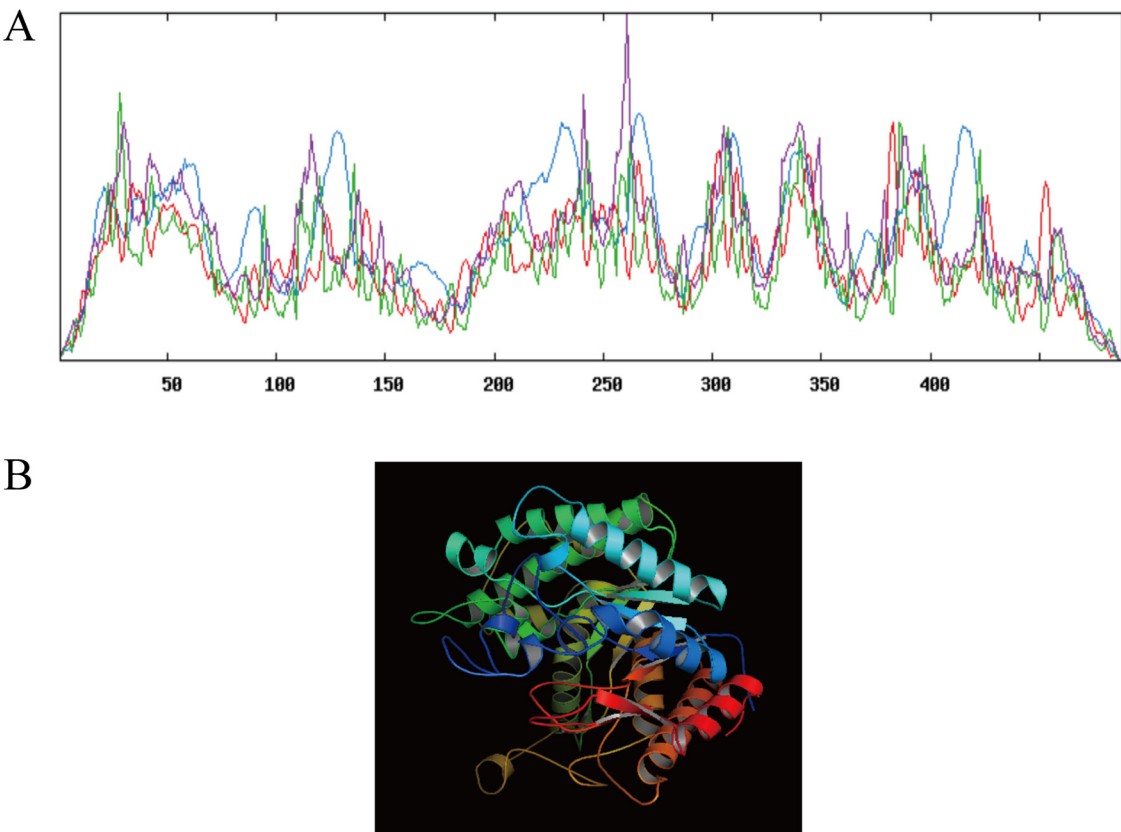

B

**FIG 5** Prediction of the structure of β-glucosidase of *B. altitudinis* JYY-02. (A) Secondary structure prediction of β-glucosidase (blue, α-helix; green, β-turn; red, extension chain; purple, random curl). (B) Tertiary structure prediction of β-glucosidase (red, N-terminal; dark blue, C-terminal).

was 1,467 bp long and encoded 488 amino acid residues, was cloned from the genome. Structural prediction of β-glucosidase from *B. altitudinis* JYY-02 showed that it contained 39.96% α-helix, which might contribute to its thermal stability. The recombinant β-glucosidase of *B. altitudinis* JYY-02 was overexpressed and easily purified. Oh et al. (20) obtained a

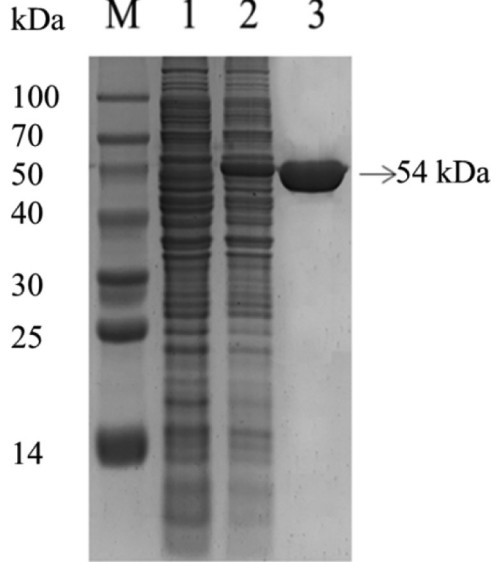

**FIG 6** SDS-PAGE analysis of the expression and purification of β-glucosidase. (Lane M) Marker; (lane 1) cell lysate of *E. coli* BL21(DE3); (lane 2) cell lysate of *E. coli* BL21(DE3) harboring plasmid pET-15b-bgl; (lane 3) recombinant β-glucosidase purified by nickel ion affinity chromatography.

**TABLE 3** Purification of recombinant β-glucosidase

| Procedure | Vol (mL) | Total protein (mg) | Total activity (U) | Sp act (U/mg) | Yield (%) |
|---|---|---|---|---|---|
| Cell lysate | 100 | 218.78 | 578.64 | 2.64 | 100 |
| Ni Sepharose 6 (FF)[a] | 20 | 45.04 | 415.20 | 9.22 | 71.75 |

[a]FF, fast flow.

yield of 0.64% for β-glucosidase by purification using various methods. In contrast, the yield of recombinant β-glucosidase in this study was 71.75% using one-step purification (20). One obvious property of the recombinant β-glucosidase was its thermal stability, with more than 60% activity remaining after treatment at 100°C for 1.5 h. Li et al. (21) purified recombinant β-glucosidase from *Dictyoglomus thermophilum* with less than 80% residual activity after holding at 75°C for 90 min. The distinct acidic optimal pH and optimal temperature were similar to those for *Aspergillus niger* (22) and *Hanseniaspora uvarum* (23).

Recombinant β-glucosidase could be used for preparation of GB by catalyzing the hydrolysis of gardenia glycoside to produce genipin, followed by the reaction of genipin with amino acids, but the reaction efficiency was relatively low, with a reaction time of 8 h at 90°C. Li et al. (11) expressed a heat-resistant β-glucosidase from *Thermotoga maritima* in *E. coli* and used it to produce GB. Single-factor and orthogonal analyses showed that exogenously expressed heat-resistant glucosidase reacted with geniposide and glycine at 94.2°C, producing a maximum yield of GB after 156.6 min. Structural adjustments of β-glucosidase by site-directed mutation of its coding gene might improve its activity and substrate specificity. Mutation studies on β-glucosidase from *Thermotoga neapolitana* led to the discovery of three mutants of β-glucosidase, which increased the hydrolytic reaction rate in an aqueous environment (24). To make possible the industrial application of the recombinant β-glucosidase of *B. altitudinis* JYY-02 for production of GB, more research is

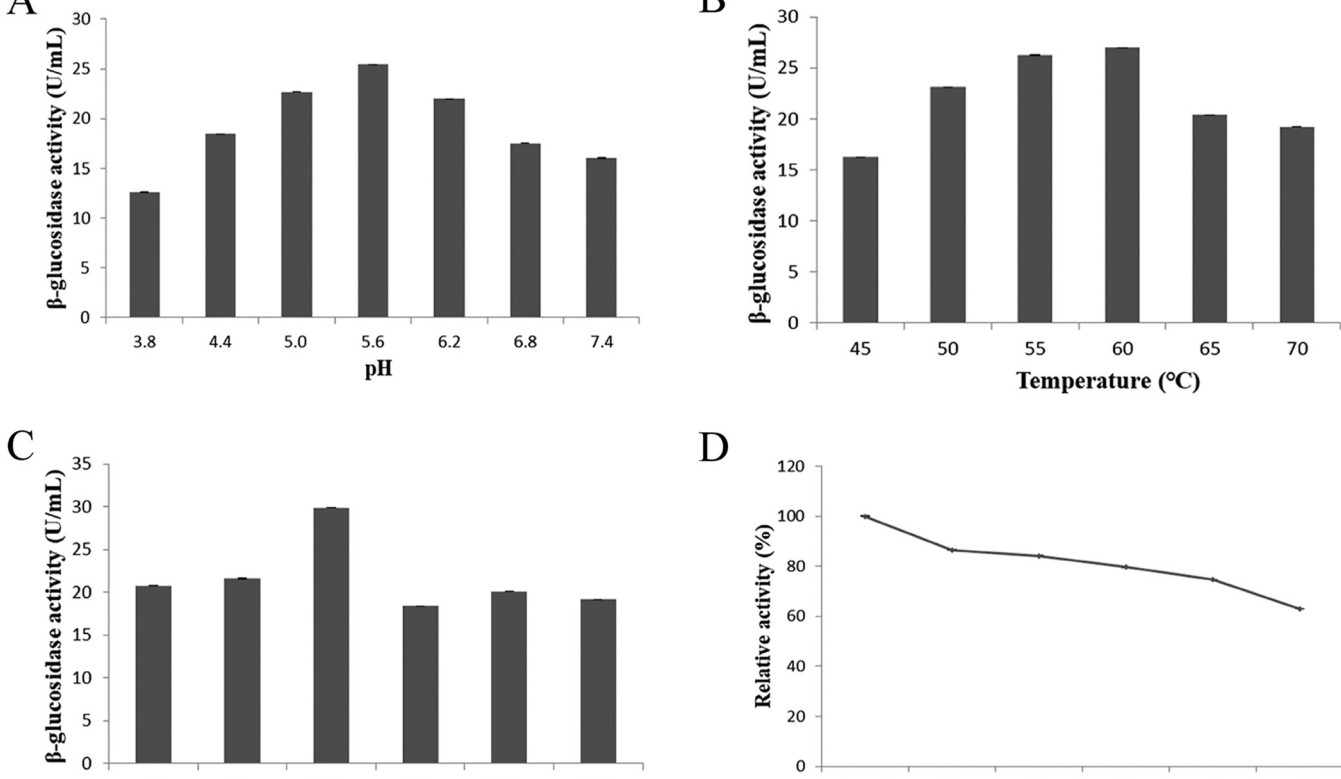

**FIG 7** Graphs showing the factors influencing the activity of recombinant β-glucosidase. Effect on enzyme activity of pH (A), temperature (B), metal ions (C), and the thermal stability of β-glucosidase (D).

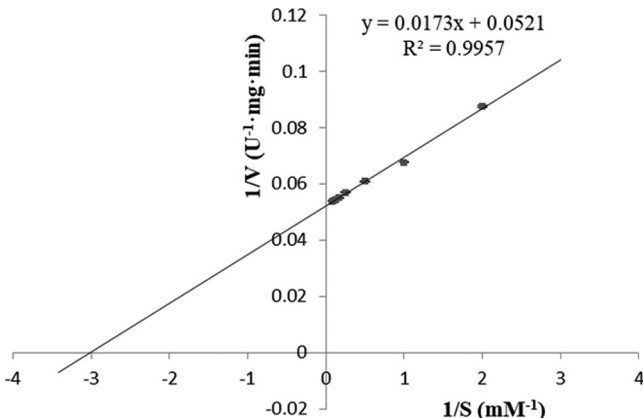

**FIG 8** Graph showing a kinetic analysis of the recombinant $\beta$-glucosidase (pNPG as the substrate).

needed to reduce production costs, for example, optimization of the reaction conditions and molecular modification of the enzyme.

In addition to glycine, a variety of amino acids can be used to prepare GB, and the kind of amino acid can also affect the color and dyeing effect of GB. Hao et al. (22) investigated the differences in the reactions of six amino acids with genipin to produce GB and found that lysine was the best among them. Besides the preparation of GB, $\beta$-glucosidase can be applied in other ways. Jo et al. (25) found that $\beta$-glucosidase-treated ginseng extracts had

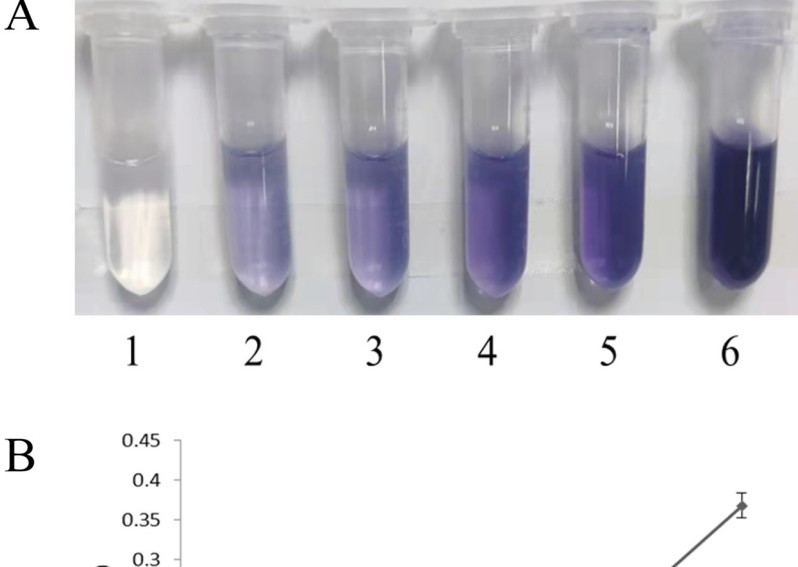

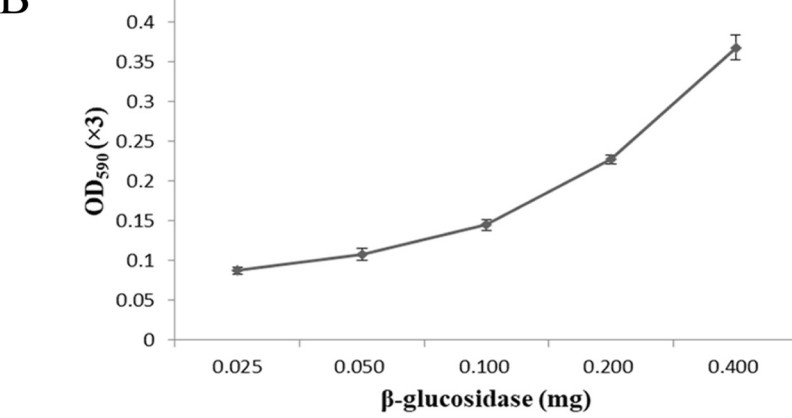

**FIG 9** Effect of the content of the recombinant $\beta$-glucosidase on GB synthesis. (A) Different enzyme amounts influencing the production of GB. Aliquots 1 through 6 were the control, 25, 50, 100, 200, and 400 $\mu$g, respectively. (B) Graph showing the relationship between the different amounts of enzymes and the relative amounts of GB.

significant cytotoxicity against cancer cells. Three β-glucosidase producing *B. subtilis* strains were reported to increase the biological activity of cheonggukjang soy grits through fermentation (26). With deeper research, new applications of β-glucosidase will be explored, which will ensure that it plays more important roles in various industrial fields.

## MATERIALS AND METHODS

**Plasmids and chemical reagents.** *E. coli* BL21(DE3) and plasmid pET-15b (Novagen, Madison, WI, USA) were used in this study. Ampicillin was purchased from Solarbio Co. Ltd. (Beijing, China). Tris, glycerol, imidazole, citric acid, NaCl, $Na_2CO_3$, $CaCl_2$, glucose, sucrose, lactose, and soluble starch were obtained from Sinopharm Chemical Reagent Co. Ltd. (Shanghai, China). The yeast extract and tryptone used in this study were produced by Oxoid (Basingstoke, Great Britain). Geniposide and p-nitrophenyl-β-D-glucopyranoside (pNPG) were purchased from Yuanye Co. Ltd. (Shanghai, China).

**Isolation and identification of strain.** Soil samples were collected from depths of 5 to 10 cm under pine trees at the campus of Qingdao University and stored in sterile sampling bags. A soil suspension was made by adding 1 g of the soil sample to 49 mL sterile water and shaking it at a speed of 150 rpm at 37°C for 4 h. The suspension was diluted by a factor of $10^6$, and then 200 $\mu$L of the suspension was coated onto Luria-Bertani (LB) solid medium (10 g tryptophan, 10 g NaCl, 5 g yeast extract, 15 g agar, and 1,000 mL distilled water, pH 7.0) containing 2.9 g/L geniposide and 1.4 g/L glycine. The plates were incubated at 37°C for 48 h, and colonies capable of producing β-glucosidase were screened out by observing the blue color surrounding them.

The four isolates secreting β-glucosidase were further purified by streaking them onto the same agar plates. Finally, an isolate named JYY-02 stably producing β-glucosidase was obtained for further study. JYY-02 was identified using morphological, biochemical, and physiological methods (27). Isolate JYY-02 was further identified by 16S rRNA gene sequencing. The 16S rRNA gene was amplified from the genomic DNA of isolate JYY-02, extracted according to the method described by Liu et al. (28). Evolutionary relationships were determined using MEGA v7.0.

**Optimization of fermentation conditions for GB production.** A single colony was incubated into a 150-mL flask containing 50 mL LB medium and incubated on a shaker at a speed of 110 rpm at 37°C until the optical density at 600 nm ($OD_{600}$) reached 0.6, and then 1.0 mL bacterial culture was inoculated into 50 mL LB liquid medium containing 2.9 g/L geniposide and 1.4 g/L glycine. The cultures were incubated at 37°C, with a shaking speed of 110 rpm and an initial pH of 7.0, for 9, 18, 27, 36, 45, 54, or 63 h to determine the optimal fermentation time. To determine the optimal initial pH for fermentation, the cultures were incubated at 37°C, with a shaking speed of 110 rpm, for 45 h with an initial pH of 6.0, 6.5, 7.0, 7.5, 8.0, 8.5, or 9.0. To check the optimal fermentation temperature, the cultures were incubated with shaking at 110 rpm, at an initial pH of 7.0, for 45 h at 25, 30, 35, 40, 45, 50, or 55°C. Different carbon sources (2%), including glucose, lactose, sucrose, and soluble starch, were added to LB liquid medium to determine the effect of different carbon sources on the fermentation production of gardenia blue, with LB liquid medium as the control. To test the optimal initial pH for fermentation, aliquots of 1.0 mL bacterial culture with an $OD_{600}$ value reaching 0.6 were inoculated into 50 mL of LB liquid medium with a pH of 5.0, 6.0, 7.0, 8.0, or 9.0. The bacteria were incubated at 37°C and 110 rpm for 45 h, and then the $OD_{600}$ value was measured. Three parallel groups were set up for each group, and the experiment was repeated three times. The OD value of the GB solution at 590 nm was used as the relative GB content. Orthogonal Design Assistant IK v3.1 was used for the design and analysis of the orthogonal experiments.

**Genome sequencing of *B. altitudinis* JYY-02.** *B. altitudinis* JYY-02 was cultured in LB medium at 45°C for 48 h in a shaker at a speed of 110 rpm, and then the culture was centrifuged at $5,000 \times g$ and 4°C for 10 min. The pellet was collected and washed with sterile water. Genome analysis was completed by BGI Genomics Co. Ltd. The genomic data of *B. altitudinis* JYY-02 were compared with those in the NCBI and CAZy databases. An amino acid sequence encoded by the β-glucosidase gene in this whole genome was highly homologous to that of a β-glucosidase gene (GenBank accession number AMM99250.1) in the NCBI database using ClustalW and ESPript v3.0 (29). After the amino acid sequence encoded by the JYY-02 β-glucosidase gene was imported into the software, secondary structure prediction of the β-glucosidases in *B. altitudinis* JYY-02 was performed using SOPMA (http://npsa-pbil.ibcp.fr/cgi-bin/npsa_automat.pl?page=npsa_sopma.html) (30), and the three dimensional structure of β-glucosidases of *B. altitudinis* JYY-02 was predicted using Phyre2 (http://www.sbg.bio.ic.ac.uk/phyre2/html/) (31).

**Expression and purification of β-glucosidase.** The β-glucosidase-encoding gene (*bgl*) was amplified from the genomic DNA by PCR amplification using two specific primers containing the BamHI and NdeI restriction sites (forward primer, AAACATATGATGACGACAATAAAAGGT; reverse primer, AAAGGATCCTCAATCAAGCGA TTCAC). The PCR product was mixed with pEASY-T1 at a 4:1 ratio and ligated overnight at 16°C to construct pEASY-T1-*bgl*. The gene *bgl* was digested from pEASY-T1-*bgl* using BamHI and NdeI and was inserted into pET-15b to construct the expression vector pET-15b-*bgl*, which was sequenced by Shanghai Biotech Biological Co. Ltd. The plasmid was transformed into *E. coli* BL21(DE3) to construct engineered bacteria.

A single colony of the engineered bacteria was inoculated into 50 mL LB medium with 100 $\mu$g/mL ampicillin and incubated at 37°C in a shaker at a speed of 110 rpm until the $OD_{600}$ value reached 0.8 to 1.0. The culture was transferred into 2.5 L LB medium with 100 $\mu$g/mL ampicillin at a ratio of 1:50 and incubated in the same way for 6 h. Isopropyl-beta-D-thiogalactopyranoside (IPTG) was then added to a final concentration of 0.5 mM, and the engineered bacteria continued to be cultured at 28°C for 6 h. A pellet was obtained by centrifugation at 10,000 rpm for 30 min at 4°C; it was suspended in 100 mL binding buffer (20 mM Tris-Cl, 0.5 M NaCl, 5.0 mM imidazole, 10% glycerol, pH 7.5). The cells were crushed in an ice bath using an ultrasonicator at

800 W for 30 cycles of 5.0 s working and 15 s idle. The cell lysate was centrifuged at 10,000 rpm and 4°C for 50 min, and the supernatant was collected for purification of the recombinant β-glucosidase.

**Purification of recombinant β-glucosidase.** The recombinant β-glucosidase was purified through affinity chromatography on a nickel column. The cell lysate supernatant was passed through a column loaded with Ni Sepharose 6 fast flow ($1.5 \times 10.0$ cm) equilibrated with binding buffer, and then the column was washed with 180 mL wash buffer (20 mM Tris-Cl, 0.5 M NaCl, 40 mM imidazole, 10% glycerol, pH 7.5). The recombinant β-glucosidase was eluted from the column using 20 mL elution buffer (20 mM Tris-Cl, 0.5 M NaCl, 0.5 M imidazole, 10% glycerol, pH 7.5) and dialyzed against TS buffer (20 mM Tris-Cl, 0.05 M NaCl, pH 7.5) for 10 h at 4°C to remove the imidazole. The homogeneity of the recombinant β-glucosidase was analyzed using SDS-PAGE, and the protein content was determined using the method of Bradford (1976) (32).

**Enzyme assay with pNPG.** A calibration curve of the $OD_{405}$ values was drawn for concentrations of p-nitrophenol solutions (0.05, 0.1, 0.2, 0.3, 0.4, and 0.5 mM) to determine the optimal content of p-nitrophenol in a reaction system. The reaction was carried out by adding 70 $\mu$L pNPG (17.86 mM) and 30 $\mu$L β-glucosidase solution (2.252 mg/mL) into 400 $\mu$L citrate-disodium hydrogen phosphate buffer (0.2 M, pH 5.6); the system was incubated at 55°C for 10 min, and then the reaction was terminated by adding 1.0 mL $Na_2CO_3$ (1.0 M). The OD values at the characteristic absorption wavelength of 405 nm were measured. The reaction system with the addition of $Na_2CO_3$ at the beginning was used as a control. The optimal pH of the recombinant β-glucosidase was measured by carrying out the reaction for 10 min at 55°C at different pH values (3.8, 4.4, 5.0, 5.6, 6.2, 6.8, and 7.4). The optimal temperature was measured by carrying out the reaction for 10 min at pH 5.6 at different temperatures (45, 50, 55, 60, 65, and 70°C). The effect of different metal ions on the activity was measured by adding $K^+$, $Zn^{2+}$, $NH_4^+$, $Ni^{2+}$, or $Mg^{2+}$ to the reaction system to a final concentration of 5 mM and carrying out the reaction for 10 min at 55°C and pH 5.6. The $OD_{405}$ of the hydrolysis product p-nitrophenol in the reaction system was detected and used as the detection index (33).

To investigate the thermal stability of β-glucosidase, the recombinant β-glucosidase solution (2.252 mg/mL) was incubated at different temperatures for 1.5 h, followed by cooling in an ice bath for 30 min, and the remaining activity was measured according to the method described above. The untreated recombinant β-glucosidase solution was used as the control. Three parallel groups were set up for each group, and the experiment was repeated three times. One unit of enzyme activity (U) was defined as the amount of enzyme which released 1 $\mu$mol of p-nitrophenol per minute under the defined reaction conditions (22). The $K_m$ value of the recombinant β-glucosidase was estimated from Lineweaver-Burk plots of data obtained with the assay solution containing pNPG at different concentrations (0.5, 1, 2, 4, 6, 8, 10, and 12 mM).

**GB synthesis with the recombinant β-glucosidase.** The recombinant β-glucosidase solution (50 $\mu$L) was added to 590 $\mu$L geniposide solution (5.0 mg/mL) in citrate-disodium hydrogen phosphate buffer (0.2 M, pH 5.6) to final contents of 25, 50, 100, 200, and 400 $\mu$g. The reaction system was incubated at 60°C for 30 min, and 290 $\mu$L glycine solution (5.0 mg/mL) was added. The subsequent reaction was carried out at 90°C for 8 h, and a clear blue color was observed. Three parallel groups were set up for each group, and the experiment was repeated three times. The $OD_{590}$ value of the reaction system was used as the relative GB content.

**Data availability.** The whole-genome sequence of *B. altitudinis* JYY-02 has been submitted to GenBank under the accession number CP093150.

## SUPPLEMENTAL MATERIAL

Supplemental material is available online only.
**SUPPLEMENTAL FILE 1**, PDF file, 1.4 MB.

## ACKNOWLEDGMENT

This work was supported by the Project for Demonstration Cities for Innovative Development in Marine Economy of Qingdao (201606).

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
