## [Reviewer comments · Microbiology Spectrum]

Microbiology Spectrum

Isolation of *Bacillus altitudinis* JYY-02 secreting a thermal-stable β -glucosidase and its application in production of gardenia blue

Jingyuan Yang, Chao Wang, Qunqun Guo, Wenjun Deng, Guicai Du, and Ronggui Li

Corresponding Author(s): Ronggui Li, Qingdao University

Review Timeline:

Submission Date:	June 3, 2022
Editorial Decision:	June 15, 2022
Revision Received:	June 26, 2022
Accepted:	June 27, 2022

Editor: Jeffrey Gralnick

Reviewer(s): The reviewers have opted to remain anonymous.

Transaction Report:

DOI: <https://doi.org/10.1128/spectrum.01535-22>

June 15, 2022

Dr. Ronggui Li
Qingdao University
308 Ningxia Road
Qingdao, Shandong 266071
China

Re: Spectrum01535-22 (Isolation of *Bacillus altitudinis* JYY-02 secreting a thermal-stable β -glucosidase and its application in production of gardenia blue)

Dear Dr. Ronggui Li:

Thank you for submitting your manuscript to Microbiology Spectrum. As you will see your paper is very close to acceptance. Please modify the manuscript along the lines I have recommended (see below). As these revisions are quite minor, I expect that you should be able to turn in the revised paper in less than 30 days, if not sooner. If your manuscript was reviewed, you will find the reviewers' comments below.

When submitting the revised version of your paper, please provide (1) point-by-point responses to the issues I raised in your cover letter, and (2) a PDF file that indicates the changes from the original submission (by highlighting or underlining the changes) as file type "Marked Up Manuscript - For Review Only". Please use this link to submit your revised manuscript. Detailed instructions on submitting your revised paper are below.

Link Not Available

Sincerely,

Jeffrey Gralnick

Comments:

- 1) Please include a 'Data Availability' statement at the end of the Methods section (please see <https://journals.asm.org/open-data-policy>). I see the information is in the Methods, but having it in its own section will help people find it more easily.
- 2) Please move Figures 5, 6 and 7 to supplemental, as this information does not seem critical to the main focus of the manuscript.
- 3) Please provide information in the methods section on how the protein structure was predicted.

Preparing Revision Guidelines

- point-by-point responses to the issues I raised in your cover letter
- Upload a compare copy of the manuscript (without figures) as a "Marked-Up Manuscript" file.
- Each figure must be uploaded as a separate file, and any multipanel figures must be assembled into one file.

- Manuscript: A .DOC version of the revised manuscript
- Figures: Editable, high-resolution, individual figure files are required at revision, TIFF or EPS files are preferred

Please return the manuscript within 60 days; if you cannot complete the modification within this time period, please contact me. If you do not wish to modify the manuscript and prefer to submit it to another journal, please notify me of your decision immediately so that the manuscript may be formally withdrawn from consideration by Microbiology Spectrum.

Dear editor,

We are sincerely grateful for the important feedback and good suggestions on manuscript “Isolation of *Bacillus altitudinis* JYY-02 secreting a thermal-stable β -glucosidase and its application in production of gardenia blue”. Based on the comments we received, careful modifications have been made on the manuscript. All of the revisions were marked in red. We hope the revised manuscript will meet the Journal’s standard.

Comment 1

Please include a 'Data Availability' statement at the end of the Methods section (please see <https://journals.asm.org/open-data-policy>). I see the information is in the Methods, but having it in its own section will help people find it more easily.

Answers: We are very grateful for the valuable comment. We have added the data feasibility section at the end of the SECTION of Materials and methods.

Comment 2

Please move Figures 5, 6 and 7 to supplemental, as this information does not seem critical to the main focus of the manuscript.

Answers: Thank you for pointing it out. We have moved Figures 5, 6, and 7 to supplemental materials and changed the serial number of figures both in Manuscript and Supplementary materials. Figures 5, 6, and 7 were changed to Figures 4S, 7S and 8S in Supplementary materials respectively, while Figures 8, 9, 10,11 and 12 were changed to 5,6,7,8 and 9 correspondingly.

Comment 3

Please provide information in the methods section on how the protein structure was predicted.

Answers: Thank you for your sincere advice. We have supplemented the websites about how protein structures are predicted in the SECTION of Materials and Methods.

Others: (1) We apologize for the valence state error of Ni²⁺ in the manuscript. We have made modifications in the MS (P7, L4; P13, L10) and Figure 7C (originally Figure 10C).

(2) The misspelling of “thoise” in P6, L1 was corrected to “those”.

(3) The sentence “The isolate was Gram-positive, with endospore and no flagella" (P4, L29) was corrected to “ The isolate was Gram-positive with endospores, and no flagellum was observed”

(4) The word ‘meaned (P6, L21)’ was corrected to ‘meant’.

(5) The sentence “ Figure 2 Phylogenetic tree of *B. altitudinis* JYY-02 (P21, L6)" was corrected to “Figure 2 Phylogenetic tree constructed from 16S rDNA sequences of *B. altitudinis* JYY-02 and other species.

June 27, 2022

Dr. Ronggui Li
Qingdao University
308 Ningxia Road
Qingdao, Shandong 266071
China

Re: Spectrum01535-22R1 (Isolation of *Bacillus altitudinis* JYY-02 secreting a thermal-stable β -glucosidase and its application in production of gardenia blue)

Dear Dr. Ronggui Li:

Your manuscript has been accepted, and I am forwarding it to the ASM Journals Department for publication. You will be notified when your proofs are ready to be viewed.

Sincerely,

Jeffrey Gralnick
Editor, Microbiology Spectrum
